# Influence of Parameters and Performance Evaluation of 3D-Printed Tungsten Mixed Filament Shields

**DOI:** 10.3390/polym14204301

**Published:** 2022-10-13

**Authors:** Myeong Seong Yoon, Hui Min Jang, Kyung Tae Kwon

**Affiliations:** 1Department of Emergency Medicine, College of Medicine, Hanyang University, 222-1, Wangsimni-ro, Seongdong-gu, Seoul 04763, Korea; 2Department of Radiological Science, Graduate School, Eulji University, 553, Sanseong-daero, Sujeong-gu, Seongnam 13135, Korea; 3Department of Radiological Science, Dongnam Health University, 50, Cheoncheon-ro 74beon-gil, Jangan-gu, Suwon 16328, Korea

**Keywords:** 3D printer, filament, shield, nozzle, X-rays, additive manufacturing

## Abstract

Currently, protective clothing used in clinical field is the most representative example of efforts to reduce radiation exposure to radiation workers. However, lead is classified as a substance harmful to the human body that can cause lead poisoning. Therefore, research on the development of lead-free radiation shielding bodies is being conducted. In this study, the shielding body was manufactured by changing the size, layer, and height of the nozzle, using a 90.7% pure tungsten filament, a 3D printer material, and we compared its performance with existing protection tools. Our findings revealed that the shielding rate of the mixed tungsten filament was higher than that of the existing protective tools, confirming its potency to replace lead as the most protective material in clinical field.

## 1. Introduction

In modern medicine, radiation provides critical information for the diagnosis and treatment of diseases, due to its ability to penetrate the human body and create visual images of the inside of the body. In medical radiation, patient exposure is not subjected to any limit because the benefit to the patient is greater than the harm caused by radiation exposure. However, repeated and continuous exposure increases the risk of stochastic radiation effects due to the absence of any threshold [1,2,3]. To prevent unnecessary radiation exposure, it is being applied to the clinical field [4,5]. Yet, there are limitations in using time and distance for radiographic examinations, and secondary exposure of organs other than the examination site may become a significant issue in the use of medical radiographic imaging equipment. Currently, in clinical practice, shielding clothing is used as the most representative example of an effort to reduce radiation exposure to workers. If the human body is exposed to strong radiation or absorbs radioactive materials that emit radiation, the cells may be damaged by the radiation and become dangerous. In addition, if the environment is contaminated with radioactive materials, ultimately humans suffer damage. Therefore, radiation shielding is necessary, and the shielding method or shielding material varies depending on the type and characteristics of radiation. In this case, lead is a metal element with high density and a high atomic number that has an excellent reaction rate and shielding power to photons [6,7] However, the use of lead shields for radiation protection is decreasing due to environmental problems and safety. Consequently, research on the development of lead-free radiation shielding bodies using ecofriendly materials is currently being conducted [8,9,10,11]. Typically, tungsten has a higher density than lead, and hence it does not react with particles and has no toxicity due to its larger attenuation coefficient. As a result, this metal is the preferred choice as a shielding material. However, tungsten is a rare metal, which is entirely imported and is difficult to process due to its high density, hardness, and melting point [12,13]. The 3D printing technology, also known as the fourth industrial revolution because of its emergence as a pioneering technology of generating physical models from digital information, can solve this problem by manufacturing shielding materials at a lower cost and can be printed out quickly with the rapid prototyping method. PLA (poly lactic acid), one of the representative materials, is an eco-friendly resin made from raw materials extracted from corn starch. It is an eco-friendly material with processability, practicality, and stability. By using the digital data of the model you want, you can easily print it anytime and anywhere at a low cost. In addition, as it is a universal technology that is not limited to specific unindustrious tries, 3D printing technology is advantageous for small-volume production of various types, such as the aerospace industry where small quantities are produced using high-quality materials or healthcare that requires customized products for each patient. There are many concrete application examples in the industry. Prior research has been reported through the method of SLS (selective laser sintering), which is one of the 3D printing technologies [14]. Studies have been conducted on the effect on the mechanical properties of polyamide components or to completely melt the powder. Nylon, a representative material of this SLS method, has excellent mechanical properties, chemical resistance, and abrasion resistance. Therefore, it is a suitable material to be applied to automobile parts and medical parts. Although the SLS method has high precision, strong physical properties, and high productivity capable of printing in large quantities, it requires user skill to use the powder (powder) form, and the printing speed is slow to print a large amount of output. It takes about 4 to 7 days. In addition, in the process of dusting off the powder, harmful microparticles are blown out, so dustproof facilities are required in the working environment. However, in the case of the FDM (fused deposition modeling) method used in this study, this process is a method of stacking the output by spraying heated filaments with an extrusion nozzle to extrude the plasticized polymer material and create the designed shape of the final product [15]. The growing interest in unit and low-volume production includes product design for specific manufacturing technologies, such as FDM and FFF, which can better exploit the advantages of these technologies and reduce their limitations [16]. The 3D printer can print quickly with a rapid prototyping method, and can be easily printed anytime and anywhere at a low cost using the digital data of the model desired by the user. The growing interest in unit and low-volume production includes product design for specific manufacturing technologies, such as FDM and FFF, which can better exploit the advantages of these technologies and reduce their limitations [17]. In addition, since the material ranges from plastic to metal, it is considered an effective substitute for developing a body shielding rate that is equivalent to that of lead. The reason is that first, pure metal has high melting point and is difficult to process, but after mixing with PLA, ABS, and so on. The melting point is relatively low and processing is easier. Second, in the case of shielding materials, the higher the atomic density, the better the shielding ability, but in the case of blended filament, the atomic density is high [18] Third, the price of pure metal 3D printer and the pure metal filament is high, and the price competitiveness of products is reduced. For this reason, metal-blended filament is used in various industries. To improve the perfection of the metal-blended filament printout, it is necessary to understand the principle of 3D printers and to be excellent in the combination of parameter settings of intrinsic factors and additional factors. However, developing a shield with optimal shielding efficiency using a 3D printer means not only understanding the principle of 3D printing, but also selecting a material with high density and shielding ability, and controlling the nozzle size, temperature, and output speed, bed temperature, Inner Fill, and Bed Height Among many of these settings, nozzle size, inner fill, and layer height are very important factors as they affect print time and quality. In general, tungsten in mineral form and filament tungsten, a 3D printer material, have different electron densities, and the filament used in 3D printers has different components depending on the manufacturer. Even if the filaments are sold under the same tungsten name, the performance of each filament varies from manufacturer to manufacturer. The reason is that tungsten + plastic (PLA) is mixed for printing with an FDM 3D printer, so the tungsten content, output parameters, and shielding rate are different for each manufacturer. In principle, the 3D printing output sets the inner fill to approximately 15–20%, as the quality of the outer surface is more important than the inner one. However, when it comes to a shielding material, the output of the inner fill is set at 100%, as the internal structure becomes critical. Greater nozzle size and layer height can allow the extrusion of more material at a given time, reducing significantly the printing time, which is very important for shielding with 100% inner fill. However, this basic 3D printer research remains insufficient. Therefore, this study produced a tungsten mixed filament shielding sheet with a content of 90.7% using FDM 3D printing technology, which has excellent processability and economic efficiency. This study aims to confirm the possibility of manufacturing effective shielding materials for X-rays and investigate the effect of nozzle size and layer height on the shielding ability.

## 2. Materials and Methods 

In this study, a shielding sheet was manufactured using a 3D printer of the FDM method using a filament mixed with Tungsten + plastic (PLA). The flow chart of the study is shown in Figure 1.

### 2.1. Using 3D Printer and Tungsten Mixed Filament Shielding Sheet Fabrication

To proceed with the experiment, the shielding sheet to be made with a 90.7% pure tungsten mixed filament was 70 × 70 mm in size and 1 mm, 2 mm, and 4 mm in thickness, using a 3D design program, Open source Free CAD 0.19. Shielding sheets were manufactured by varying the size and layer height.

### 2.2. Setting the Output Variable and Converting the G-Code

The 3D model was converted to a STL file and then sliced to multiple layers (Cura 4.9.1, Ultimaker, S3 The Netherlands), as shown in Table 1. The nozzle size was set to 0.4 and 0.8 mm, the layer height was set to 0.1, 0.2, 0.3, and 0.4 mm, respectively, and the output speed, nozzle temperature, bed temperature, and internal filling were set, as shown in Table 1. The G-code generated for printing was sent to the 3D printer using the network (Table 1).

### 2.3. The 3D Printing Output and Shielding Sheet Production

The transmitted G-code was transmitted to a 3D printer to print a shielding sheet. In case of Figure 2a, it is a 3D printer (Ultimaker, S3 The Netherlands) used in the experiment, and in the case of Figure 2b, it is a shielding sheet produced through G-code transmitted to the 3D printer.

## 3. X-ray Dosimetry

### 3.1. Dosimetry in X-rays

To assess the shielding performance of the shielding sheet produced using a tungsten mixed filament with a purity of 90.7%, the distance between the X-ray source and the shielding sheet was 1000 mm, and the distance between the shielding sheet and the ionizing box for dosimetry was 50 mm from the center of the general imaging field. The shielding sheet made of tungsten blend filament was changed to 1 mm, 2 mm, 3 mm, 4 mm, and 5 mm, respectively, and an X-ray generator (DR System, GX-650H, DK Medical Solutions, Korea) and an ionization chamber for dosimetry (Rad-check plus, Fluke, United States of America) was used to detect the dose. The average dose was calculated by measuring each dose 10 times.

### 3.2. Shielding Rate Test Using Shielding Sheet

To evaluate the effect of nozzle size and layer height on the shielding sheet produced using a tungsten filament and a 3D printer, we performed the same experimental process as the lead equivalent test method of the Korean industry standard X-ray protective products. A diagnostic radiation generator (DR System, GX-650H, Dong Kang, Korea) and an irradiation dosimeter (Rad-check plus, Fluke, United States of America) were used as the experimental equipment. Before the experiment, the reproducibility of the diagnostic radiation generator was assessed, and the equipment’s good condition performance was confirmed.

The shielding rate was calculated by applying Equation (1) to the measured irradiation dose [unit: mR].
(1)$$Shielding\;Rate\;\left(\%\right)=\frac{\left(W_{0}\right.-\left.W\right)}{W_{0}}\times 100$$
where *W* and *W*_0_ represent the radiation dose in the presence and absence of a shielding sheet, respectively.

### 3.3. Dosimetry of Lead Protective Tools

To compare the difference between the shielding sheet made of tungsten and the conventional lead protection tool, the X-ray test was additionally performed. The irradiation dose and shielding rate of an apron (0.175 mmPb, 0.25 mmPb, and 0.35 mmPb), and a thyroid protector were measured and compared with the shielding rate of the shielding sheet.

## 4. Results

### 4.1. Dosimetry of X-rays

#### 4.1.1. Shielding Sheet Printing Time

Table 2 shows the printing time of the shielding sheet produced by the 3D printer. Measured at bed heights of 0.1 mm, 0.2 mm, 0.3 mm, and 0.4 mm on a 0.4 mm nozzle, the printing times were 7 h 8 min, 3 h 45 min, 2 h 30 min, and 1 h 58 min, respectively. Furthermore, the printing time with the 0.8 mm nozzle was 2 h 32 min at a bed height of 0.1 mm, 1 h 20 min at 0.2 mm, 54 min at 0.3 mm, and 42 min at 0.4 mm (Table 2).

#### 4.1.2. Evaluating the Measured Dose and Shielding Rate

Table 3 shows the results of dose measurement and shielding rate using a shielding sheet made of tungsten filament by changing the nozzle size and layer height of the 3D printer. At 100 kVp, 40mAs, and shielding, the 0.4 mm nozzle showed a shielding rate of 92.23% at a height of 0.4 mm, and a 0.8 mm nozzle under the same conditions showed a shielding rate of 93.49% at a height of 0.4 mm (Table 3).

#### 4.1.3. X-ray Dose Measurement and Evaluation of Shielding Rate

The X-ray dose measurements and shielding rates were as follows (Table 4): 7.33 mR for 1 mm tungsten filament at 120 kVp, 20 mAs, 93.52% shielding. For 5 mm tungsten filament, the 0mR shielding rate is 100% at 120 kvp 20, 40 mAs.

#### 4.1.4. Comparison of the Shielding Rate between Lead Protective Tools and Shielding Sheets

The results of the dose and shielding rate measurements of the lead protective device were as follows (Table 5): 133.67 mR dose in the absence of a shielding sheet, at 120 kVp, 20 mAs; 32.33 mR for the 0.175 mmPb shielding sheet, shielding rate of 74.51%; 19.33 mR for the 0.25 mmPb sheet, shielding rate of 85.64%; 14.33 mR for the 0.35 mmPb sheet plus the thyroid protector, shielding rate of 89.28%. Furthermore, the dose was 274.33 mR in the absence of a shielding sheet at 120 kVp, 40 mAs; 63.67 mR for 0.175 mmPb, shielding rate of 76.49%; 19.33 mR for 0.25 mmPb, shielding rate of 85.64%, 28.67 mR for 0.35 mmPb, shielding rate of 89.75%. The thyroid protector showed a shielding rate of 90.09%, at a dose of 27.67 mR. Compared to the tungsten shielding sheet, at 120 kVp and 20 mAs: 7.33 mR dose for tungsten 1 mm, shielding rate of 93.52%; 1.67 mR for tungsten 2 mm, shielding rate of 98.05%; 0.33 mR for tungsten 3 mm, shielding rate of 98.75%; 0 mR for tungsten 4 mm and 5 mm, shielding rate of 100 %. At 120 kVp, 40 mAs: tungsten 1 mm was 94.00% at 13.67 mR; tungsten 2 mm was 3.67 mR, 98.64%; tungsten 3 mm was 0.67 mR, 99.81%; tungsten 4 mm was 0.33 mR, 99.88%. Figure 3 demonstrates that at a shielding rate of 0 mR, 100% was obtained for tungsten 5 mm, verifying a better shielding effect than a lead protective tool (Table 5).

## 5. Discussion

Because of its high density, lead is very effective in reducing radiation exposure, making it the most valuable material in the X-ray department [11]. However, lead is toxic to the human body, and if lead enters the blood, there is a high risk of lead poisoning [11,19,20,21]. Therefore, research is being conducted, focusing on the development of lead-free shielding materials. Additive manufacturing (AM) is an innovative approach to industrial production that allows the creation of 3D products by layering different materials, such as plastics, metals, or concrete, to create lighter and stronger parts and systems. However, it is no longer used solely to produce prototypes and is slowly establishing itself as a sequential production method in the automotive, aerospace, medical and sports industries. Currently, 3D printing technology is being applied in various ways. According to previous studies, it has been reported that it can be applied to education and research, orthoses and implants in the medical field, surgery, and surgery to solve individual problems with individual customization and low cost [13,22,23]. Additionally, although there are countless applications of 3D printing, one area where more research is being done and which appears to be continuously growing is the medical field. In radiology, several imaging processes are used to capture 2D images of the human body, but the most common techniques are CT and MRI. CT can represent the internal structure of the cross-section as an image by reconstructing the attenuation data passing through the subject using a mathematical technique. High-resolution image acquisition and bone structure, as well as quality control of exercise organs, such as the cardiovascular system, are possible [24]. With the current software development, 3D modeling is possible using continuous DICOM format CT images, and it has become easy to output modeling data to a 3D printer of the same size, but materials that can accurately represent human organs are still lacking. In addition, the material cost of the filament is high, and the filament used in 3D printers has a wide range of materials to choose from. However, it is necessary to select a material suitable for the application because the density and physical properties are different depending on the filament used [25]. According to the results of another previous study by Laurynas Gilys et al., 69.5 keV is the value for W and 90.5 keV is the value for Bi; all these metals have a higher X-ray absorption capacity than pure lead (K-absorbing edge-88 keV) at certain energy intervals and were reported to cover the entire medical diagnosis range [26]. Therefore, this study used tungsten mixed filaments with a density of 7.5 g/cm^3^, and 3D printing technology, to determine the optimal nozzle size, the amount of material extrusion, and the height at which materials are stacked layer by layer. Shielding bodies with equivalent shielding performance were the output, and differences were compared and analyzed through shielding experiments. Our experimental process revealed that the time required to print a shape with the same size while using the same equipment and the same material to print the shield body, was 428 min, and at least 42 min depending on the set value. In addition, pertaining to the nozzle size, a 0.8 mm nozzle that could extrude a larger material width was 60 kVp, 1.21%, 80 kVp, 1.08%, 100 kVp, 1.23%, suggesting that the dose shielding rate is superior to that of the 0.4 mm nozzle. Furthermore, our findings showed that a lower layer height in the 0.4 mm nozzle and a larger layer height in the 0.8 mm nozzle demonstrate better dose shielding rate. We then evaluated the dose and shielding rate of tungsten mixed filament shielding sheets, namely tungsten, apron 0.175, 0.25, 0.35 mmPb, and thyroid protector, through X-rays. Our results showed that the shielding sheet made of tungsten mixed filament had more than 90% shielding rate from 1 mm. Compared with the existing lead protection tools, 0.175 mmPb, 0.25 mmPb, 0.35 mmPb, and thyroid protector all showed lower shielding rates, compared to tungsten mixed shielding sheets, confirming the possibility that this material can be a potential lead substitute. According to Yin Wu’s research, using tungsten powder as a reinforcing agent, a novel gamma-ray shielding material was fabricated through the FDM 3D printing process and reported that it had the ability to effectively shield low-energy γ-rays. Based on these prior studies, tungsten is preferred as a shielding material that can replace lead [27]. In general, the filament used in 3D printers consists of different components that vary according to the manufacturer, and hence its performance is different as well. In addition, there are insufficient experiments to fabricate a shield using tungsten mixed filaments, and most of the existing studies conducted experiments with simulations. Therefore, the tungsten content of the filament used in this study reached 90.7%, and it was confirmed that the performance did not deteriorate through comparative evaluation with lead protection tools that are actually used in clinical field using tungsten mixed filaments. The 3D printing is considered to influence the quality of the output product depending on the material properties required for lamination or lamination of bends. As the nozzle size becomes smaller, a filament mixed with a metal material may clog or wear the nozzle hole due to metal impurities in the filament. Therefore, the filament manufacturer recommends that the nozzle size should be 0.6 mm or larger when using a metal mixed filament. Considering the shielding performance, printing time, and filament manufacturer’s recommendations, our study revealed that a 0.8 mm nozzle size is better than a 0.4 mm nozzle, both in terms of shielding performance and 3D printer mechanical aspects. These results are expected to be based on the ideal additional factor parameters when fabricating shields use a 3D printer. However, experimental studies on the development of a shielding body using tungsten-blended filaments are insufficient, and tungsten, the material of the shielding sheet, is not flexible. Therefore, research needs to focus on the manufacturing of a flexible shielding sheet that is properly mixed with other materials. Consequently, the properties of this shielding sheet will render this material a lead protection tool, allowing its efficient application in various clinical fields.

## 6. Conclusions

This study assessed the effect of nozzle size and layer height in the manufacturing of a shielding body using tungsten mixed filament and 3D printing. Our findings verify that the shielding rate could be increased by up to 1.23% while reducing the printing time by 1/10 depending on the nozzle size and the height of the layer. It is believed that these results can provide a greater insight into the production of efficient shielding sheets with 3D printing technology. We believe that the manufactured tungsten shield can replace the existing protective tools used in clinical practice, and future studies can render this material suitable for use in nuclear medicine and therapeutic settings.

## Figures and Tables

**Figure 1 polymers-14-04301-f001:**
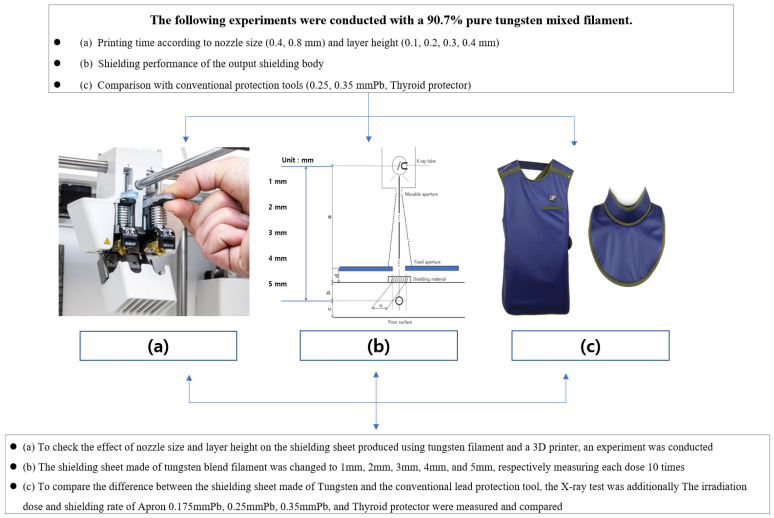
Flowchart of this study. (**a**) In the process of manufacturing and printing a shielding sheet using a 3D printer and a tungsten mixed filament, the change in the shielding ratio according to the noz-zle size, layer, and height is evaluated. (**b**) In the case of shielding rate, it was tested through the lead equivalent test method of the Korean industry-standard X-ray protective products. (**c**) A comparative analysis was conducted using Apron and thyroid protectors used in clinical practice to confirm the possibility that the printed shielding sheet can replace the existing protective tools.

**Figure 2 polymers-14-04301-f002:**
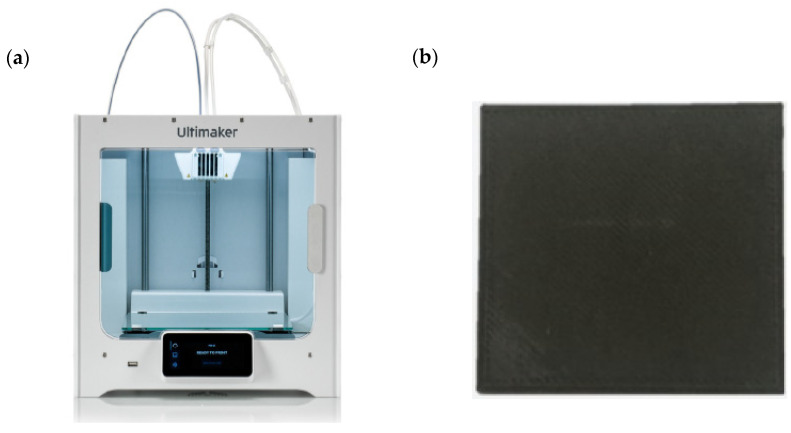
(**a**) The shielding sheet required for the experiment was output using a 3D printer. (**b**) A shielding sheet was manufactured using a tungsten mixed filament (Tungsten + PLA).

**Figure 3 polymers-14-04301-f003:**
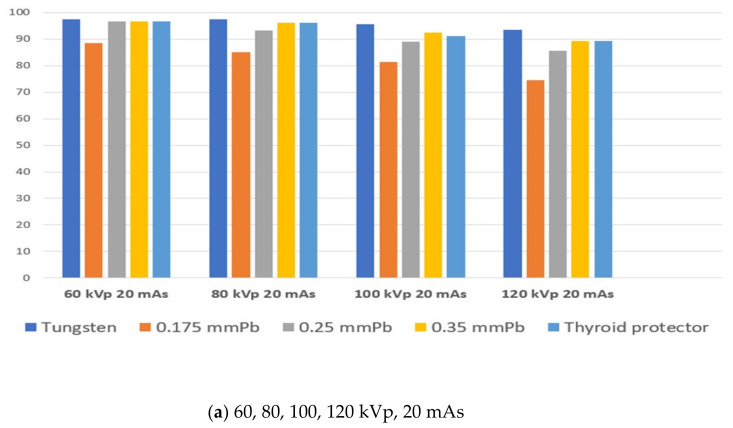
Comparative evaluation of the manufactured shielding sheet and lead protection tool. Based on 1 mm of the manufactured tungsten shield, (**a**) 60, 80, 100, 120 kVp 20 mAs and (**b**) 60, 80, 100, 120 kVp 40 mAs were evaluated for comparison with the existing protection tools.

**Table 1 polymers-14-04301-t001:** The 3D printer settings.

	Radiation Shield Sheet
Nozzle size (mm)	0. 4, 0.8
Layer Height (mm)	0.1, 0.2, 0.3, 0.4
Printing temperature (°C)	213
Bed temperature (°C)	65
Infill Density (%)	100
Printing speed (mm/s)	50

**Table 2 polymers-14-04301-t002:** Shielding sheet manufacturing time using a 3D printer.

Nozzle Size	0.4	0.8
Layer Height		
0.1 mm	428 min	152 min
0.2 mm	225 min	80 min
0.3 mm	150 min	54 min
0.4 mm	118 min	42 min

**Table 3 polymers-14-04301-t003:** Results of radiation shielding sheet tests.

-		60 kVp, 40 mAS		80 kVp, 40 mAs		100 kVp, 40 mAs	
Nozzle Size	Layer Height	Dose [mR]	Shielding Rate [%]	Dose [mR]	Shielding Rate [%]	Dose [mR]	Shielding Rate [%]
non		123.8	-	212.4	-	326.6	-
nozzle 0.4 mm	0.1 mm	3.0	97.66	8.6	94.94	23.1	91.81
	0.2 mm	1.6	98.75	8.8	94.86	23.8	91.61
	0.3 mm	1.6	98.75	9.4	94.37	24.6	92.07
	0.4 mm	1.6	98.75	9.8	94.19	25.1	92.23
nozzle 0.8 mm	0.1 mm	1.5	98.79	8.75	94.88	22	93.26
	0.2 mm	1.5	98.79	8.25	96.12	21.5	93
	0.3 mm	1.5	98.79	8.25	96.12	21.25	93.49
	0.4 mm	1.5	98.79	7.5	96.57	21.25	93.49

**Table 4 polymers-14-04301-t004:** Results of the X-ray radiation shielding sheet tests (60, 80, 100, 120 kVp, 20, 40 mAs).

X-rays												
Sheet	kVp	mAs	1 mm		2 mm		3 mm		4 mm		5 mm	
			Dose [mR]	Shield Ring rate [%]	Dose [mR]	Shielding Rate [%]	Dose [mR]	Shielding Rate [%]	Dose [mR]	Shielding Rate [%]	Dose [mR]	Shielding Rate [%]
Tungsten	60	20	0.67	97.38	0	100	0	100	0	100	0	100
		40	0.67	98.17	0	100	0	100	0	100	0	100
	80	20	1.67	97.52	0.33	96.51	0	100	0	100	0	100
		40	3.67	96.29	1.33	97.88	0	100	0	100	0	100
	100	20	3.33	95.58	1.33	98.05	0	100	0	100	0	100
		40	8.33	94.80	1.67	98.91	0.33	98.83	0	100	0	100
	120	20	7.33	93.52	1.67	98.05	0.33	98.75	0	100	0	100
		40	13.67	94.00	3.67	98.64	0.67	99.81	0.33	99.88	0	100

**Table 5 polymers-14-04301-t005:** X-ray results for radiation apron, thyroid protector tests (60, 80, 100, 120 kVp, 20, 40 mAs).

kVp	mAs	None	0.175 mmPb		0.25 mmPb		0.35 mmPb		Thyroid Protector	
		Dose [mR]	Dose [mR]	Shielding Rate [%]	Dose [mR]	Shielding Rate [%]	Dose [mR]	Shielding Rate [%]	Dose [mR]	Shielding Rate [%]
60	20	41.33	4.33	88.52	1.33	96.58	1.33	96.68	1.33	96.78
	40	81.00	7.67	89.53	1.33	98.01	1.33	98.06	1.33	98.56
80	20	67.33	9.33	85.14	4.33	93.33	2.33	96.24	2.67	96.05
	40	135.67	18.67	86.24	8.33	93.96	4.33	97.11	5.33	96.57
100	20	97.33	17.67	81.35	10.67	89.04	6.67	92.45	7.67	91.12
	40	198.33	37.67	80.61	21.33	89.44	13.33	93.08	15.33	92.57
120	20	133.67	32.33	74.51	19.33	85.64	14.33	89.28	14.33	89.28
	40	274.33	63.67	76.49	38.33	87.13	28.67	89.75	27.67	90.09

## Data Availability

Not applicable.

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
