# Peer review of "Influence of Parameters and Performance Evaluation of 3D-Printed Tungsten Mixed Filament Shields"

_polymers, 2022, doi:10.3390/polym14204301_

Round 1

Reviewer 1 Report

This effort of authors is much appreciated towards finding the effect of parameters on performance of 3D printed tungsten mixed filaments. However, there is much room to improve this manuscript. 

1. Novelity is highly unclear.

2. Research Methodology is full of jargons specifically the terms shielding rate, tungsten mixing with PLA, use of unclear units like mR, etc. 

3. Results and Discussion contain considerable jargons too. 

4. Quite often spelling mistakes e.g. uesing instead of using for Fig. 3.

5. There is no need of extensive repeat of sentences in Discussion section.

Author Response

1. There are many different methods for shielding, but prior research is lacking in manufacturing a shield body using a 3D printer. In addition, studies comparing lead and lead using Tungsten are also lacking. Therefore, it is believed that this study will be very helpful for related research.

2. mR means radiation dose in SI standard units just like the word CM is used.
Without talking about this, the ability of shielding cannot be evaluated.

3. As the expertise of radiation and the expertise of polymers are combined, it seems that it should be mentioned little by little. I tried to explain it in an easy way as the reviewer said.
We ask for your understanding in the remaining parts for the completeness of this thesis.

4. Correction has been completed

5. I edited it as you said.

What you said has been edited in red.

More details are attached

please check.

Reviewer 2 Report

The subject is interesting and I propose it for publication by considering the following highlights:

1. I propose to consider adding a few sentences regarding the explanation of additive manufacturing and the most important techniques. you can consider the following reference: https://doi.org/10.3390/polym14173674

2. Figure 1: the quality of the figure is not good. I propose to increase the quality of the entire image.

3. Table 1: The reference should be mentioned, or the datasheet that has been adapted from.

4. Figure 3: The quality is not good, it should be modified.

Author Response

As people who majored in radiology and are currently working in the clinic, we are doing a lot of research to shield and prevent unnecessary radiation from being exposed to patients.

Although there are many studies on shielding materials with 3d printers, it is thought that there is no more efficient method than this with FDM method. Confirming that the shielding rate increases or decreases according to the setting value while using the same material is asserting the scientific basis that we must now consider the setting method of the 3D printer.

This study may seem insufficient, but for us, if this study is approved, in future studies, we will conduct advanced research that meets the needs of reviewers.

1. I added a sentence by referencing it through the link you referenced.
2. I have improved the quality of the picture, please check it
3. Modified with reference
4. The picture has also been modified

Round 2

Reviewer 1 Report

Discussion may further be improved leading to its more expansion.

The novelty paragraph must be put at the end of introduction section and before materials and methods section.

Thanks

Author Response

  1. Discussion may further be improved leading to its expansion

->  As the reviewer said, we have added the content of the discussion part.

  1.  The novelty paragraph must be put at the end of the introduction section and before the materials and methods section

->  I've corrected it based on what you said.

As people who majored in radiology and are currently working in the clinic, we are doing a lot of research to shield and prevent unnecessary radiation from being exposed to patients.

Although there are many studies on shielding materials with 3d printers, it is thought that there is no more efficient method than this with FDM method. Confirming that the shielding rate increases or decreases according to the setting value while using the same material is asserting the scientific basis that we must now consider the setting method of the 3D printer.

This study may seem insufficient, but for us, if this study is approved, in future studies, we will conduct advanced research that meets the needs of reviewer
